# Selection and Incorporation of siRNA Carrying Non-Viral Vector for Sustained Delivery from Gellan Gum Hydrogels

**DOI:** 10.3390/pharmaceutics13101546

**Published:** 2021-09-23

**Authors:** Anastasios Nalbadis, Marie-Luise Trutschel, Henrike Lucas, Jana Luetzkendorf, Annette Meister, Karsten Mäder

**Affiliations:** 1Department of Pharmaceutical Technology, Faculty of Natural Sciences 1-Biosciences, Martin Luther University Halle-Wittenberg, 06120 Halle/Saale, Germany; anastasios.nalbadis@pharmazie.uni-halle.de (A.N.); marie-luise.trutschel@pharmazie.uni-halle.de (M.-L.T.); henrike.lucas@pharmazie.uni-halle.de (H.L.); 2Department of Internal Medicine IV (Oncology/Hematology), Faculty of Medicine, Martin Luther University Halle-Wittenberg, 06120 Halle/Saale, Germany; jana.luetzkendorf@uk-halle.de; 3ZIK HALOmem and Institute of Biochemistry and Biotechnology, Faculty of Natural Sciences 1-Biosciences, Martin Luther University Halle-Wittenberg, 06120 Halle/Saale, Germany; annette.meister@chemie.uni-halle.de

**Keywords:** siRNA, gene delivery, local controlled release, hydrogel, non-viral, lipoplex

## Abstract

The local controlled release of siRNA is an attractive and rational strategy to enhance and extend the effectiveness of gene therapy. Since naked and unmodified siRNA has a limited cell uptake and knockdown efficiency, the complexation of siRNA with non-viral carriers is often necessary for the delivery of bioactive RNA. We evaluated the performance of three different non-viral siRNA carriers, including DOTAP lipoplexes (DL), chitosan polyplexes (CP), and solid lipid complexes (SLC). The physicochemical properties of the siRNA-nanocarriers were characterized by dynamic light scattering and gel electrophoresis. After in vitro characterization, the carrier with the most appropriate properties was found to be the DL suspension, which was subsequently loaded into a gellan gum hydrogel matrix and examined for its drug load, stability, and homogeneity. The hydrogels microstructure was investigated by rheology to assess the impact of the rheological properties on the release of the siRNA nanocarriers. A controlled release of complexed siRNA over 60 days in vitro was observed. By comparing the results from fluorescence imaging with data received from HPLC measurements, fluorescence imaging was found to be an appropriate tool to measure the release of siRNA complexes. Finally, the bioactivity of the siRNA released from hydrogel was tested and compared to free DL for its ability to knockdown the GFP expression in a DLD1 colon cancer cell model. The results indicate controlled release properties and activity of the released siRNA. In conclusion, the developed formulation is a promising system to provide local controlled release of siRNA over several weeks.

## 1. Introduction

After the discovery of RNA interference in 1998 by Fire et al. [1], the downregulation of proteins became a valuable approach for the treatment of various diseases [2]. Even though gene knockdown offers plenty of medical applications, siRNA-mediated gene knockdown is limited by the delivery to the specific location, due to challenging pharmacokinetic properties of siRNA [3]. siRNA is a highly hydrophilic macromolecule that is quickly degraded by endogenous nucleases and renally eliminated. In addition, the cellular uptake of free siRNA is low [4].

To overcome these barriers, non-viral complexing carriers have been explored to facilitate siRNA transfection. Complexation with cationic compounds such as lipids or polymers can protect bound siRNA from nucleases, improve cell uptake, and transport siRNA to targeted tissues and cells [5].

Based on the work of Felgner et al. [6], the complexation with a cationic lipid, whose cationic head group interacts with the negatively charged phosphate backbone results in higher stability and protection from nucleases. The overall cationic charge of the colloidal carrier associates with the negatively charged cell membrane followed by cellular uptake through endocytosis [7]. The delivery of siRNA chitosan polyplexes was first introduced by Howard et al. [8]. Chitosan, a naturally occurring polyglucosamine, is capable of forming associates with siRNA due to the cationic charge of the primary amino group. Due to its interesting attributes, such as quick cell internalization, biocompatibility, and mucoadhesivity, chitosan gained a lot of attention for gene delivery [9]. Furthermore, chitosan polyplexes offer a high structural versatility with tunable properties, which made chitosan an interesting gene carrier for many purposes such as intravenous, intranasal, intratumoral, pulmonary, and even, oral delivery [10,11,12,13]. Solid lipid nanoparticles are alternative carrier systems that consist, in contrast to lipoplexes, of a solid lipid core (fatty acids, glycerides, waxes). Solid lipid nanoparticle complexes (SLC) for siRNA delivery, include also cationic compounds, that associate with the negatively charged phosphate backbone of siRNAs and thus enable siRNA complexation. Among other applications, siRNA-loaded SLC were investigated for nose-to-brain delivery against Alzheimer’s disease [14], for the treatment of liver disorders [15], and for cancer therapy [16].

Unfortunately, siRNA-induced knockdown effects have only a transient effect on cells and, therefore, the therapeutic effect will decrease after a short time. The short-time activity is a serious limitation for further therapeutical use of siRNA. A high number of repeated administrations is an undesirable scenario to solve this problem. A rational and highly attractive alternative is the development of drug delivery systems, which provide a local and controlled release (CR-DDS). Local controlled release may not only provide a long-lasting effect but also minimize toxicity, which may appear when applied systemically. Examples of CR-DDS include microparticles [17], hydrogels [18,19], scaffolds [20], layer-by-layer films [21] as well as nano-fibers [22]. Hydrogels are water-swollen, three-dimensional networks that consist of natural or synthetic polymers with high water-binding efficiency. Since they imitate native tissue, hydrogels are not only being used for providing a drug depot, but also for tissue engineering purposes [23]. Saito et al. [24] used gelatin hydrogels for the controlled release of bioactive siRNA. The formulation degraded within 40 h, thus releasing the incorporated complexes very quickly. Longer release times were achieved by Schwabe et al. [25], using hydrogels for the release of polyethylenimine-siRNA polyplexes.

We focused our efforts on the development of hydrogels for controlled siRNA delivery with respect to a high translational potential. Therefore, already pharmaceutically used hydrogel-forming excipients with an excellent safety profile are of primary interest. Gellan gum is an FDA-approved biomaterial and is treated as a future-oriented candidate material in biomedical engineering. Besides its uses for wound dressing, bone regeneration material, and further applications in biomedicine, gellan gum is used as drug delivery vehicle [26]. It is a linear negatively charged polysaccharide that forms cross-linked hydrogels in the presence of ions. Due to low cytotoxicity, those hydrogels are highly biocompatible [27] and can be used for ophthalmic delivery [28]. Goyal et al. [29] developed gellan gum PEI-DNA-siRNA nanocomposites, in which gellan gum improved bioactivity in vitro and after intravenous injection, in vivo. Due to these properties, gellan gum offers a wide range of applicability as therapeutic drug depots. The hydrogels are stable over a large pH range, physiologically compatible, not cytotoxic and show good shear thinning properties allowing the injection in a syringe [30]. Gellan gum also offers in situ gelling behavior [28], increasing the therapeutic potential even further. Besides the external administration for ocular, dermal, or vaginal diseases, the injectability would allow the application for subcutaneous, intra-articular, or intratumoral indications.

Due to these promising attributes, our study aimed to develop a siRNA-loaded gellan gum hydrogel for local and controlled release applications. The development can be separated into three parts in which the carrier finding and characterization is part I, the drug loading and hydrogel formulation is part II and the controlled release is part III.

## 2. Materials and Methods

### 2.1. Materials

The following: 2-(4-(2-hydroxyethyl)-1-piperazinyl)-ethanesulfonic acid (HEPES) buffer, dichloromethane, Dulbecco’s phosphate-buffered saline (PBS), RPMI 1640 cell culture medium, Opti MEM™ fetal bovine serum, trypsin-EDTA, Triton™ X100 and dibutylamine, were obtained from Sigma-Aldrich, (St. Louis, Missouri, MO, USA). Cetylpyridinium chloride, acetonitrile HPLC-grade, and glacial acetic acid HPLC-grade were purchased from VWR (Radnor, Pennsylvania, PA, USA). 1,2 dioleoyl-3 dimethylammonium-propane (DOTAP) was purchased from Avanti^®^ Polar Lipids, Inc. (Alabaster, Alabama AL, USA). Chitosan lactate was obtained from Heppe Medical Chitosan (Halle/Saale, Germany) and cetyl palmitate (Kollicream^®^ CP 15) was bought from BASF (Ludwigshafen, Germany). Gellan gum Kelcogel^®^ GG-LA was bought from CP-Kelco^®^ (Atlanta, Georgia, GA, USA).

Lipofectamine™ 2000 transfection reagent, Silencer™ select Cy3-labeled Negative Control No. 1 siRNA, and Silencer™ GFP (eGFP) siRNA were obtained from Life Technologies (Carlsbad, California, CA, USA). Allstars negative control siRNA was purchased from Qiagen (Venlo, Nederlands).

GFP expressing DLD 1 colon carcinoma cell line (ATCC CCL-221™) was kindly provided by the working group of Dr. Thomas Müller (Department for Internal medicine IV, Universitätsklinikum Halle).

### 2.2. Lipoplex, Chitosan Polyplex and Solid Lipid Complex Preparation

The DOTAP lipoplex (DL) was prepared by the thin-film hydration method. An amount of 10 mg of DOTAP was dissolved in methylene chloride and subsequently evaporated in a vacuum overnight. The dried film was hydrated with 1 mL sterile (HEPES) buffer (pH 7.4) forming large multilamellar bilayer structures with a final lipid concentration of 10 mg/mL. The suspension was thoroughly vortexed and extruded through polycarbonate membranes (pore size 100 nm) at room temperature. The stock solution was stored at 6 °C until usage.

Chitosan polyplexes (CP) were prepared freshly by dissolving the polymer in HEPES buffer overnight and subsequent mixing with siRNA.

For the solid lipid complexes (SLC), 4% (*m*/*V*) cetyl palmitate was molten at 60 °C, mixed with 1% (*m*/*V*) cationic lipid cetylpyridinium chloride and produced by a hot emulsion technique in 100 mL water with a high-pressure homogenizer (Emulsiflex C5, Avestin Inc., Ottawa, Canada). The emulsion underwent 5 cycles at a pressure of 1000 bar with subsequent cooling in an ice bath. The SLC were stored at 6 °C until usage.

Before the experiments, the nanocarrier was thoroughly mixed in the required ratio with siRNA and incubated for 20 min at room temperature. For cell experiments, the complexing agent and siRNA were diluted in Opti-MEM^TM^ medium, mixed, and incubated for 20 min at room temperature before their administration on the cells.

### 2.3. Size and Zeta Potential Measurements

The measurements were executed on a Zetasizer Nano ZS (Malvern Panalytical, Worcestershire, UK) using the backscattering mode.

The DL suspension was diluted 1:10, CP was measured at a concentration of 0.1% (*m*/*V*) and the SLC was diluted 1:50 in filtered double distilled water. For particle diameters, each sample was measured in triplicate with 15 sub runs and analyzed intensity-weighted. The equilibration time of the device was set to 2 min. All measurements were performed in the attenuator range from 6 to 9. For zeta-potential measurements, the sub-runs were set automatically by the device. The data were analyzed by the manufacturer‘s software (Zetasizer version 6.30).

### 2.4. Agarose Gel Electrophoresis

The N/P ratio required for complexation was determined by agarose gel electrophoresis. A total of 4% agarose gels were loaded with 0.3 µg/mL ethidium bromide solution (final concentration) in 1% TAE buffer (pH 8). The complexes were prepared and incubated for 20 min before they were mixed with 10 µL of glycerol/water solution (50% *v*/*v*). The electrophoresis was conducted at 200 V for 1 h. Afterward, the gel was removed from the electrophoresis unit and pictures of the fluorescent siRNA bands were taken in a UVP UVsolo Touch imager (Analytik Jena GmbH, Jena, Germany). The gels were analyzed by the manufacturer’s software (VisionWorks, version 4.15).

### 2.5. Toxicity Studies

5000 DLD1 cells per well (ATCC CCL-221) were seeded in 48 wells of a 96 well-plate and grown for 24 h at 37 °C and 5% CO_2_ in 100 µL RPMI Medium enriched with 10% (*v*/*v*) fetal bovine serum (FBS) and 1% (*v*/*v*) penicillin/streptomycin. In each well, 100 µL of nanocarriers was added in the respective concentration. The cells were incubated for up to 96 h and the cell viability was determined by a resazurin reduction assay.

For this purpose, 30 µL of a 0.15 mg/mL resazurin solution was added to the cells and the mixture was incubated for 2 h. The fluorescence intensity was measured by a Cytation^TM^ imaging reader (BioTek Instruments Inc, Winooski, Vermont, VT, USA). The excitation wavelength was set to 531 nm, the emission wavelength to 593 nm.

The cell viability was determined after subtraction of the blank (empty wells) and by setting the negative control (untreated cells) to 100%. A total of 0% cell viability was assessed by Triton™ X100 treated cells. All measurements were performed in triplicate.

### 2.6. Gel Fabrication

An amount of 10 mg low acetylated gellan gum was dissolved in 0.8 mL of double-distilled RNAse free water at 70 °C in a water bath. The complex-siRNA suspension was mixed and incubated in a total volume of 100 µL for 20 min to ensure complete complexation and afterward added to the gellan gum sol. An amount of 100 µL of 10% (*m*/*V*) sodium chloride solution was added and mixed. Before gelation, the sol was evenly distributed into a round mold forming thin gel plates. For complete gelation, the gel was incubated overnight at 6 °C.

### 2.7. HPLC Analysis for Stability and Release Experiments

Stability assessment after gel fabrication was performed by ion-pair high-performance liquid chromatography (HPLC) analysis in combination with UV detection.

The hydrogels were incubated in PBS in a shaker under light protection at 37 °C. After 24 h, the medium was taken for the measurement. The samples were measured with an Agilent 1100 series HPLC system (Agilent Technologies Inc. Santa Clara, California, USA) equipped with a YMC Triart C18 (150 mm × 3 mm; 5 µm) column (YMC Europe, Dinslaken, Germany).

The HPLC method was carried out at 50 °C by gradient elution with mobile phase A containing 1 L water, 4 mL dibutylamine, and 1.4 mL glacial acetic acid. Mobile phase B consisted of 50% mobile phase A and 50% acetonitrile. The flow rate was set to 1 mL/min and the injection volume to 10 µL. The gradient started with 25% mobile phase B and increased up to 100% mobile phase B at minute 20. Until minute 25, the method ran out with 100% mobile phase A. Stable siRNA was eluted with a retention time of 14.5 min and detected at a wavelength of 260 nm.

### 2.8. Rheology

Rheological measurements were executed on a Malvern Kinexus Lab + rheometer (Malvern Panalytical GmbH, Nürnberg, Germany). The measurements were carried out in a cone-plate geometry with a plate diameter of 40 mm and a cone angle of 4° in oscillatory mode at 37 °C The liquid soles were poured in a mold and after gelation loaded on the geometry.

To determine the linear viscoelastic region (LVR), the hydrogels were measured with oscillatory amplitude sweep (shear strain 0.1–10%, frequency 1 Hz). For frequency sweep measurements, the shear strain was set to 0.3% and frequency was altered from 0.1 to 100 Hz.

The structural parameters were calculated on the basis of the rubber elastic theory (RET) from the following equations:(1)$$\xi =\left(\frac{{\;G\prime \;N}_{A}}{\mathrm{RT}}\right)$$

ξ is the mesh size (nm), G′ is the storage modulus, N_A_ is the Avogadro constant (6.022 × 10^23^), R is the gas constant (8.314 J/K mol) and T is the temperature (in K).

The crosslinking density (n_e_) was calculated as follows:(2)$$n_{e}=\frac{G_{e}}{\mathrm{RT}}$$

G_e_ is the storage modulus at the plateau phase in frequency sweep measurements.

### 2.9. Complex Distribution and Release Kinetics

For visualization of siRNA-complex distribution, the gels were loaded with complexed fluorescence-labeled siRNA-Cy3, and measurements were carried out using the Maestro^TM^ fluorescence imaging system (CRi, Massachusetts, MA, USA) and the Maestro software (version 2.10). A green filter set was used in the range of 550 to 650 nm in 5 nm steps using automatic exposure times. Emission spectra as well as the autofluorescence signal of an unloaded hydrogel as background were recorded. For release experiments, the gels were incubated in 800 µL of PBS in a shaker under light protection at 37 °C. At regular time intervals of 2 days, the medium was taken for the measurements and replaced with fresh buffer solution afterward. As a bleaching control, non-incubated siRNA-Cy3 loaded hydrogels were measured in the same time intervals.

### 2.10. Carrier and Hydrogel GFP Knockdown

200.000 GFP-DLD1 cells per well were seeded in 6 well plates and grown for 24 h at 37 °C and 5% CO_2_ in 2 mL RPMI Medium (10% FCS, 1% P/S). After cell adhesion, the medium was changed to 2 mL RPMI Medium (0% FCS, 0% P/S). For carrier transfection, the complexing components were incubated in Opti-MEM™ medium for 20 min before application.

For gel production, all solutions were filtrated through a 0.2 µm pore size filter and produced in a sterile environment. The lipoplex-loaded hydrogels were prepared and incubated for 24 h in Opti-MEM™ medium. After 24 h, the release medium was applied on the cells.

Ninety-six hours after transfection, the cells were washed and harvested in tubes for GFP quantification by fluorescence-activated cell sorting (FACS) analysis. The GFP expression was determined by setting the negative control (negative control siRNA-treated) to 100%. FACS analysis was performed on a FACSCalibur (BD Biosciences, San Jose, Californa, CA, USA) with CellQuest software. For each measurement, 10,000 events were counted. The data was analyzed by gating living cells from cell debris in the first step and by doublet discrimination in the second step.

### 2.11. Statistical Analysis

Data were calculated as mean. The standard deviation is shown by the brackets. Statistics were calculated with *t*-tests. A *p*-value of ≤ 0.05 was considered significant.

## 3. Results and Discussion

### 3.1. Carrier Characterization

Table 1 lists the values of the size distribution (z-average, PDI) and the charge properties (zeta potential) of the investigated siRNA carriers.

The lipid-containing DL carrier has a monomodal size distribution with a low PDI and hydrodynamic diameter of 130.0 ± 0.9 nm. The zeta potential is the highest of the screened carriers.

The chitosan-based complex CP is smaller at 50.0 ± 20.2 nm, but shows a significantly higher PDI of 0.290, which is explained by the heterogeneity of chitosan polymers.

Solid lipid-containing carriers SLC have a similar size as the DL suspensions with a hydrodynamic diameter of 144.9 ± 0.3 nm and a low PDI. The particle sizes, as well as the particle morphology, was approved by transmission electron microscopy, (Appendix A). The zeta potential of SLC is the lowest among the three examined carriers. Determination of the required N/P ratio was performed with agarose gel electrophoresis for the respective carrier (Figure 1). For CP, the band intensity in the starting point increases with increasing concentration of chitosan, while band intensity decreases at the height of siRNA due to the improved retention. The siRNA bands remain visible at all N/P ratios. At a ratio of 20:1 complexation was complete. The DL suspension is capable to completely bind siRNA at a N/P ratio of 1:1. In contrast to CP, the band intensities at the starting point disappear due to a higher condensation efficiency, which prevents the intercalation of ethidium bromide with siRNA [31].

For the SLC, a N/P ratio of 10:1 was necessary for complexation. At higher ratios of 50:1 and higher, the condensation was sufficient to inhibit the fluorescence emission. On basis of these results, the N/P ratio for further experiments was set to 1:1 for DL, 20:1 for CP and 10:1 for the SLC.

### 3.2. Carrier Cytotoxicity

Non-viral siRNA carriers are promising vehicles to overcome barriers faced by siRNA delivery. However, those are limited by their cytotoxicity in vitro and in vivo [32]. To compare the cellular toxicity of the different siRNA carriers, human colorectal adenocarcinoma cells (DLD1) were investigated 96 h after application of 0.1 µg/mL, 1 µg/mL, and 3 µg/mL of complexed siRNA. A graphical representation of cytotoxicity is shown in Figure 2. After exposition to DL, the DLD1 cells showed no significant acute toxicity upon transfection at all concentrations and varied only slightly from the vital control group.

The CP was tolerated as well by the DLD1 cells and showed no acute toxicity, even at the highest concentration, which is a manifold of the used concentration (0.54 µg/mL) for transfection experiments. The DLD1 cells responded to the application of SLC with a decreased viability comparable to the level of the negative control group at all applied concentrations.

Based on the results from the cell viability experiments, the SLC was discontinued for further investigation as a possible gene carrier.

### 3.3. Carrier GFP Knockdown Efficiency

The DL and CP were examined for their ability to transfect GFP-expressing DLD1 cells (Figure 3A). The DL suspension was able to decrease the GFP signal significantly to 59.3% (*t*-test, *p* < 0.05); however, CP reduced the GFP expression to 95.3%.

CP was not as toxic as the SLC but performed poorly compared to DL in transfection experiments. Even though chitosan is reported to have sufficient transfection capabilities, it fails to deliver convincing results in the used DLD1 cell model. Parameters such as molecular weight [33] and N/P ratio [34] can be modified to achieve better results. Due to the higher knockdown efficiency, the DL suspension was chosen as the best performing carrier for the loading on a controlled release depot formulation.

### 3.4. Hydrogel Formulation: siRNA Recovery, Stability and Lipoplex Distribution

The hydrogels are exemplary, shown in Figure 3B in bright light and Figure 3C excitated at a wavelength of 550 nm. The hydrogels have smooth surfaces and did not show any signs of erosion during the experimental observation time of 10 weeks. The fluorescence intensity is equal all over the hydrogel, indicating the lipoplex-siRNACy3 to be distributed homogeneously. For stability assessment, the chromatograms (Figure 4) of blank received from unloaded hydrogel, untreated siRNA, released siRNA, and degraded siRNA were compared after incubation in PBS at 37 °C for 24 h. A total of 99.03% ± 0.01% of the loaded siRNA was recovered from the loaded hydrogel after the production proving high stability of the double-stranded siRNA during the gelation process.

Since short 21-nucleotide RNA lacks a secondary structure, only one peak (after 14.6 min) is visible, which is typical for stable double-stranded siRNA [35]. Smaller peaks with lower retention times are visible in the chromatogram of the degraded siRNA at 8.5 min and 11 min. These peaks result from shorter siRNA degradation products, that, with increasing amounts of acetonitrile, are desorbed from the column faster than larger fragments [36] and, therefore, elute first. Other peaks are degradation products from the gellan gum hydrogel, which appear also in chromatograms of a blank hydrogel.

### 3.5. Oscillation Rheology: Determination of Mesh Size

Besides the ionic interactions between the network and the lipoplex that sustains the release, diffusion through the network is one of the main factors for the release from hydrogels [37]. The networks mesh size significantly affects the release rate with which hydrogels release APIs when used as drug delivery systems [38]. Therefore, it is important to understand the hydrogel’s microstructure.

The hydrogels were examined for the rheological behavior by oscillation rheology. First, the linear-viscoelastic region (LVR) was determined by an amplitude sweep method at a constant frequency of 1 Hz starting at a shear strain of 0.1% to 10% (Figure 5).

G′, which represents the elastic properties of the gels, was increasing with higher concentrated gels. All formulations showed a linear rheological behavior from 0.2% to 0.8% shear strain. At 0.8% shear strain, G′ was decreasing while G″ and the δ increased, indicating the irreversible destruction of the network structure [38].

For all formulations in the LVR, G′ was higher than G″. δ was measured between 0° and 10°. Due to these rheological properties, the hydrogels can be characterized as strong and stable.

Concluding from the amplitude sweep measurements, frequency sweep measurements were performed at 0.3% shear strain within the LVR (Figure 6). The frequency range was set from 0.1 Hz to 100 Hz. G′ was increased with higher concentrations of gelator. With increasing frequency, G′ remained constant up to a frequency of 60 Hz. After this plateau phase, G′ increased at higher frequencies. This is caused by polymer chains in the network that are not able to recede at high frequencies [39].

The G′ from the plateau can be used to determine the average mesh size (ξ) and the crosslinking density (N_ε_) of the hydrogels on the basis of rubber elastic theory (RET). ξ is defined as the average distance of two junction points in the network and N_ε_ is defined as the number of crosslinks in the network per specific volume.

Table 2 lists the calculated results for the hydrogels. The average mesh size ξ decreased in higher concentrated hydrogels from 17.0 nm (0.5% gellan gum) down to 7.9 nm (at 2% gellan gum) concentration. The calculated cross-linking density N_ε_ increased tenfold when the gellan gum concentration was increased from 0.5% to 2%.

According to the results, higher concentrated hydrogels have a smaller average mesh size, a higher density of crosslinks and are, therefore, expected to release the lipoplex slower, and vice versa, due to the denser physical barriers. Doubling the concentration from 0.5% to 1% had a stronger impact on the change in the mesh size and the cross-linking density compared to the moderate changes observed after doubling the gellan gum concentration from 1% to 2%.

### 3.6. Release from the Hydrogel

Release experiments were performed with naked siRNA-Cy3 or lipoplex-siRNA-Cy3 with different hydrogel concentrations. The results are presented in Figure 7. After 48 h, the release of naked siRNA from the network was completed. A very long release time over 2 months of DL was achieved at the highest hydrogel concentration when still 31% of initial fluorescence emission was measured after 63 days. The further release was expected for longer observation. The release of the lipoplexes from the formulation with 1% gellan gum was completed after 1 month. All formulations showed a burst release within the first 24 h. After 24 h, a slower release of the lipoplexes becomes visible.

The release of siRNA was either controlled by complexation or by the concentration of the gelling agent. Small hydrophilic siRNA is released quickly from the network due to its small molecular size and hydrophilic properties. The negatively loaded phosphate backbone of the siRNA does not interact with the negatively loaded hydrogel resulting in no further retention [40].

Different reasons are hypothesized to be responsible for the extensive prolonged release compared to naked siRNA. First, the lipoplex carries an overall positive charge and, therefore, is expected to interact with the negatively charged gellan gum network [41].

Second, lipoplexes are larger structures that are hold back by the physical hydrogel network. The denser the hydrogel network—the more the release was slowed down. Since no degradation was observable and is reported to be slow for gellan gum [41], the release mechanism of the hydrogels is assumed to be dominated by diffusion through the network. The release curves of the complex formulations show two different release phases, in which the burst release matches the release of the 1% naked siRNA formulation, independent of the gel concentration. We hypothesized that during this phase, naked decomplexed siRNA is released first at the same rate because it is largely independent of the gel concentration. Anionic polymers, such as gellan gum, are capable to release bound siRNA from cationic complexes [42] In the second phase, the release is slowed down due to the larger size of the lipoplexes, depending on the gel concentration. The difference in release kinetics indicates two different analytes eluting from the drug depot. First, naked siRNA is quickly released during the burst release, followed by the slower release of complexed siRNA.

### 3.7. Release Comparison Analysis

The measurement of released lipoplexes from depot formulations is challenging. We decided to use two methods (HPLC and fluorescence) for independent quantification of the released siRNA to verify that the release of siRNA-Cy3 is correctly recorded by fluorescence detection. The detection of emission is sensitive enough for quantitative measurements but might be potentially prone to errors. Figure 8 presents the releases of both HPLC and fluorescence-imager measured experiments.

The release start and endpoint, as well as the release rate, are correctly recorded by both methods. The value of released siRNA determined by HPLC reaches almost 100% of the loaded siRNA-Cy3 after 24 h; however, the results obtained with the fluorescence imager indicate 12% remaining fluorescence intensity at the same time. The fluorescence intensity dropped to 9% after a further 24 h. The remaining signal can be explained by autofluorescence effects of the gellan gum or fluorescence interferences caused by minor environmental changes in each measurement. With decreasing fluorescence intensity during the experiment, the measurement requires higher sensitivity, which is gained by longer exposure times resulting in higher background noise. Due to the high sensitivity of fluorescence imaging to interferences, the standard deviation of the release data is larger than the standard deviation from HPLC measurements. In conclusion, the release of siRNA-Cy3 can be appropriately tracked by fluorescence imaging, which was used to measure the release of siRNA-Cy3 lipoplexes.

### 3.8. Hydrogel GFP Knockdown Efficiency

The produced DL as well as the hydrogel formulation were tested for their transfection efficiency. To monitor the specific knockdown, the stable GFP expressing colon carcinoma cell line DLD1 was selected. The knockdown efficacy is shown by the difference between the negative control-treated cells (set to 100% GFP expression) and the lipoplex and hydrogel-treated cells (Figure 9). The free lipoplexes knocked down the gene expression to a statistically significant 65%, compared to the negative control-treated cells. As observed in the histogram received from FACS analysis, the population treated with free lipoplexes was shifted to lower fluorescence intensities flattening the received curve and shifting the maximum by the power of ten.

The hydrogel formulated lipoplexes slightly reduced the GFP level to 94.4% ± 3.3%. Compared to free lipoplexes, the lower knockdown efficiency was expected for the following reasons.

First, the cumulative application of the drug for cell transfection is an artificial approach, which only partially represents in vivo conditions, e.g., direct cell contact on the depot surface. Possible optimization of transfection efficiency from depot formulations is incorporating the cells into [37] the depot formulation or laying the cells on the formulation [25] for a direct contact cell adhesion on the hydrogel approach. The second limitation is the faster drug deactivation of siRNA complexes during the prolonged incubation in cell culture sera [43]. Third, the released dose within 24 h is about 30% of the actual drug load. Higher efficiencies with higher doses can be expected. Additionally, as already mentioned in Section 3.6, we hypothesized the initial burst release to be naked siRNA that could be decomplexed by the surrounding polyelectrolyte in the drug depot, which could also explain the lower effect.

## 4. Conclusions

In this work, we describe the development of a siRNA delivery depot. We evaluated three different siRNA complexing carriers (DL, CP, SLC) for their potential as non-viral gene delivery vehicles. Based on the results, the SLC carrier was found to be too toxic and, therefore, was discontinued. In GFP knockdown experiments, the DL suspension was the best performing delivery carrier and, therefore, loaded into a drug depot, a gellan gum hydrogel formulation. The formulation was investigated for drug loading, siRNA stability, and homogeneity. The recovery of stable siRNA was close to 100%, which proves the successful loading into the hydrogel network. Homogeneity within the formulation was assessed by fluorescence imaging, implying that siRNA-Cy3 complexes were evenly distributed all over the hydrogel. Due to the slow degradation of gellan gum, diffusion of the network is most likely the primary release mechanism. Therefore, the microstructure of the hydrogel was investigated by oscillation rheology. The rheology measurements reveal that the crosslink density increases tenfold, and the pore size is reduced by more than 50% in the range from 0.5% to 2%, thus slowing down diffusion-controlled release.

In release experiments, the formulation was able to sustain the release, dependent on the hydrogel concentration, as well as through complexation. While naked siRNA is quickly released within 24 h, a maximum release of lipoplex-siRNA of 60 days was achieved. Due to the different release kinetics, we concluded that in the burst release time interval, unretained naked siRNA is mostly released. In the second release phase, complexed siRNA in the form of lipoplexes is slowly released.

To assess whether fluorescence imaging delivers appropriate results for the release, the method was compared to the release measured by HPLC analysis. The results gained from both methods were comparable and, therefore, also applicable for tracking the release of siRNA in a lipoplex-bound state.

Free lipoplexes, as well as hydrogel-embedded lipoplexes, were examined for their transfection efficiency. Lipoplexes in suspension could significantly reduce the GFP expression of DLD1 cells. Even though hydrogels were able to reduce protein synthesis in the cells, the effect was lowered compared to the free lipoplexes. This may be caused by the slower release of carriers in the experiment time frame of 24 h, at which time point the lipoplex released should be at a maximum of 30% of the loaded lipoplexes.

Future experiments will focus on the optimization of cell transfection and the control of drug release. We could sustain the release of siRNA over 60 days, which now provides the basis for a long-lasting release of bioactive siRNA, bringing the therapeutic application of siRNA depot formulations one step closer to its intended in vivo use.

## Figures and Tables

**Figure 1 pharmaceutics-13-01546-f001:**
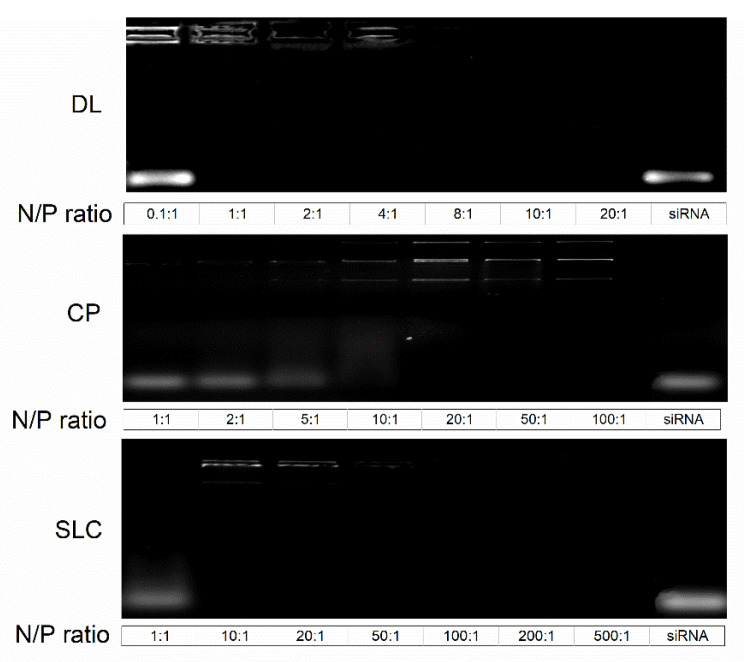
Analysis of siRNA loading efficiency in the respective N/P ratio for the respective nanocarriers. The last lane represents uncomplexed siRNA.

**Figure 2 pharmaceutics-13-01546-f002:**
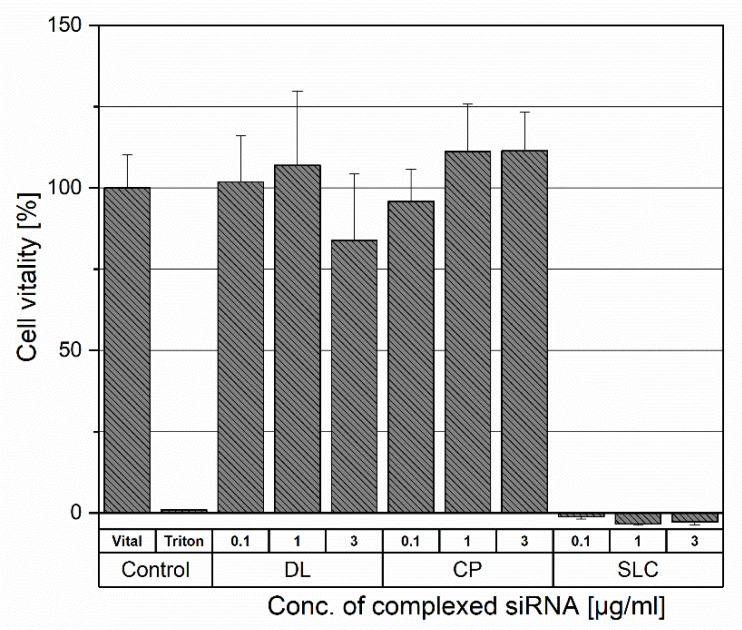
Cytotoxicity of DL, CP, and SLC 96 h after transfection at concentrations of 0.1, 1 and 3 µg/mL of complexed siRNA. The control group consists of the untreated cells for the 100% vitality value and the Triton X100 treated group for the maximum cytotoxic value.

**Figure 3 pharmaceutics-13-01546-f003:**
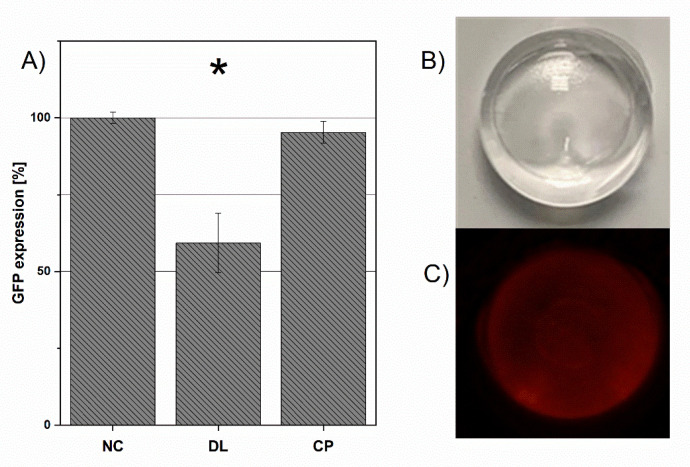
(**A**) represents the GFP expression level of the negative control (NC) group, DL and CP. Significant results (*p* < 0.5) is marked by *. Figure 3B,C show lipoplex-siRNACy3 loaded hydrogels in bright light (**B**) and excitated at a wavelength of 550 nm (**C**). Fluorescence imaging was used to assess the lipoplex-siRNACy3 distribution in the hydrogel (Figure 3B). After loading, the siRNA-Cy3 emission is homogeneously distributed.

**Figure 4 pharmaceutics-13-01546-f004:**
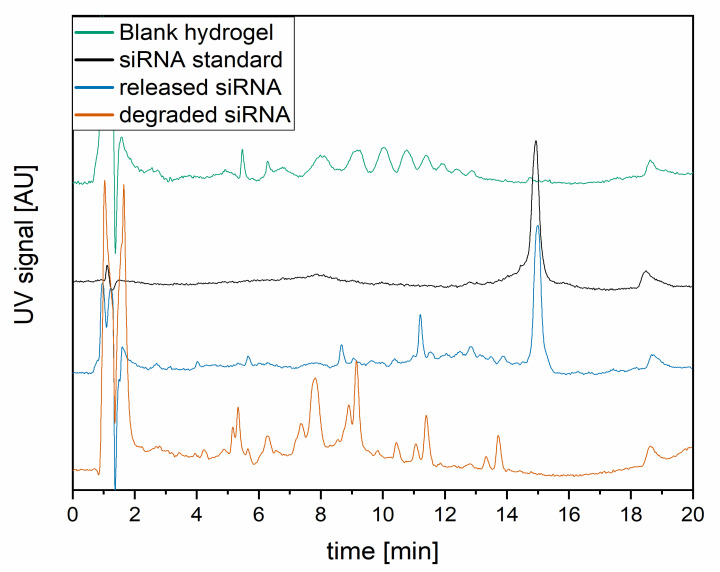
HPLC chromatograms from stability experiments of loaded siRNA in hydrogels (**blue**), compared to the siRNA standard (**black**), blank from unloaded hydrogel formulation (**green**), and degraded siRNA (**orange**).

**Figure 5 pharmaceutics-13-01546-f005:**
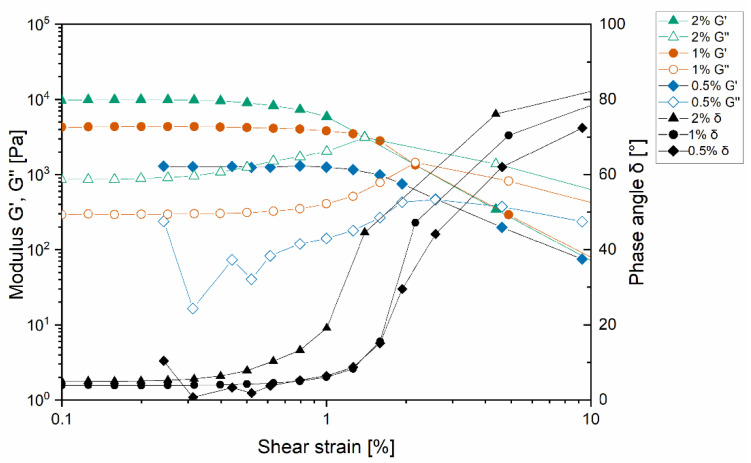
Amplitude sweep measurement for the definition of the linear viscoelastic region of the hydrogels. 0.5% is represented in squares, 1% in circles, and 2% in triangles. Storage modulus, loss modulus, and phase angle are shown for each concentration.

**Figure 6 pharmaceutics-13-01546-f006:**
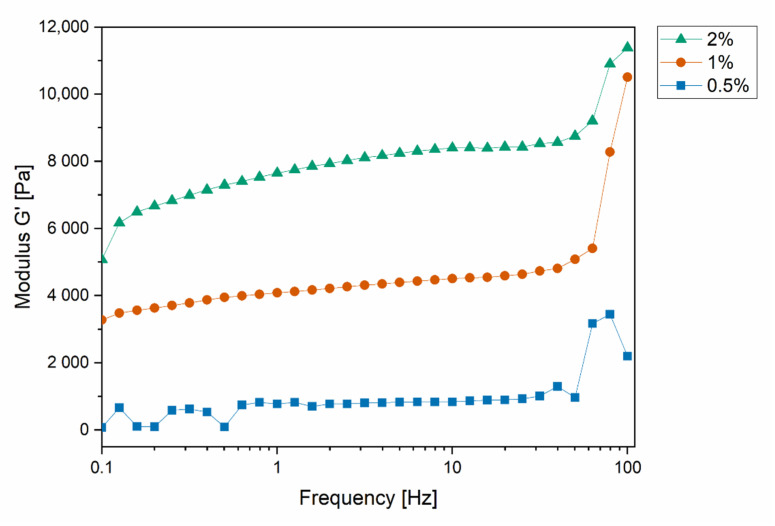
Storage modulus received from frequency sweep measurement with a set shear strain of 0.3% for determination of the storage modulus at the plateau phase.

**Figure 7 pharmaceutics-13-01546-f007:**
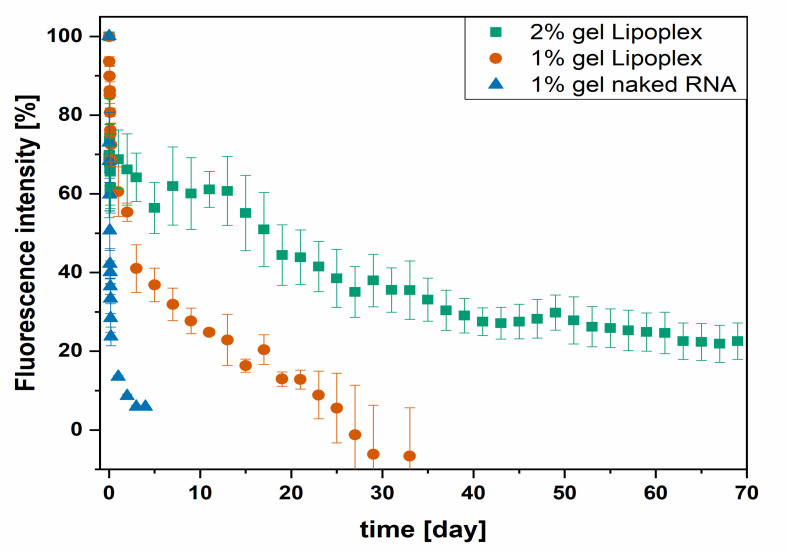
Release of either naked siRNA or lipoplex from the formulation of 1% and 2% gelator concentration.

**Figure 8 pharmaceutics-13-01546-f008:**
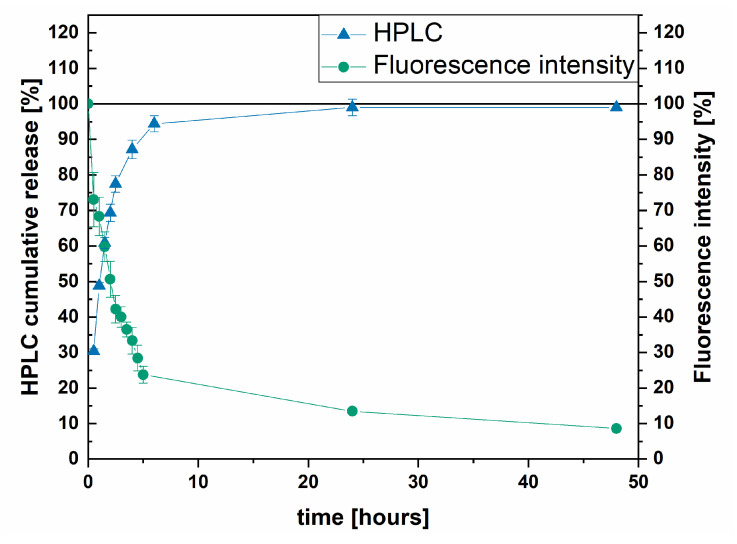
Release of siRNA-Cy3 measured by fluorescence imaging and HPLC.

**Figure 9 pharmaceutics-13-01546-f009:**
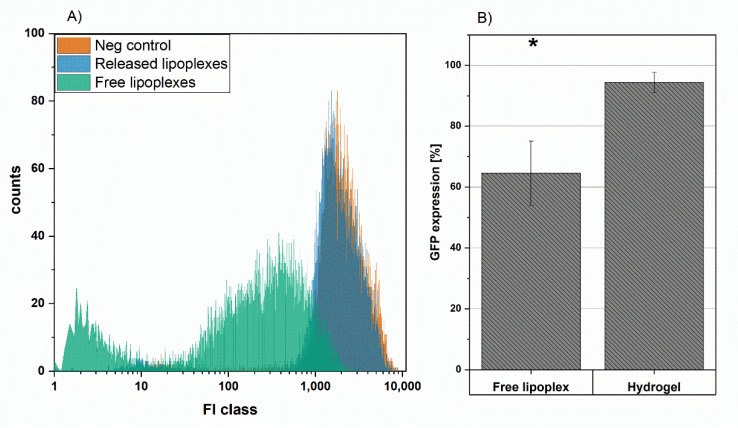
(**A**) FACS intensity histogram and (**B**) resulting mean GFP knockdown of free lipoplexes compared to hydrogel released carrier, the negative controls represent 100% of the GFP expression. * corresponds to a statistical significance of *p* < 0.05.

**Table 1 pharmaceutics-13-01546-t001:** Size, size distribution and zeta potential measurements of DL, CP, and SLC.

	Z-Average (nm) ± SD	PDI	Zeta Potential (*mV*) ± SD
DL	130.0 ± 0.9	0.078	78.3 ± 13.3
CP	50.0 ± 20.2	0.290	21.4 ± 3.4
SLC	144.9 ± 0.3	0.110	7.6 ± 0.7

**Table 2 pharmaceutics-13-01546-t002:** Storage (G′) and loss moduli (G″) received from frequency sweep measurement and calculated from these measurements, the mesh size (ξ) and crosslink density (N_ε_).

Gel Conc. (*m*/*V*) [%]	G′ [Pa]	G″ [Pa]	ξ [nm]	N_ε_ [mol/m^3^]
0.5	840.3	53.08	17.0	0.339
1	4506	229.9	9.70	1.818
2	8402	513.7	7.88	3.390

## Data Availability

Not applicable.

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
