# Peer review of "Selection and Incorporation of siRNA Carrying Non-Viral Vector for Sustained Delivery from Gellan Gum Hydrogels"

_pharmaceutics, 2021, doi:10.3390/pharmaceutics13101546_

Round 1

Reviewer 1 Report

The manuscript evaluates the performances of three different non-viral siRNA carriers, DOTAP lipoplexes (DL), chitosan polyplexes (CP) and solid lipid complexes (SLC). The authors prepared siRNA carriers and evaluated their sizes, toxicity, mesh size, hydrogel formation and release utilizing widely used techniques in the same field. They also measured to deliver the siRNAs inside cells and to see their effects on GFP knockdown efficiency. They found SLC carrier to be toxic. They report that DL suspension as the most promising carrier for the loading on a controlled release depot formulation due to its higher transfection efficiency.

  • The manuscript is well planned, executed and written.
  • The author presented results very efficiently.
  • The resolution of figure 1 may be improved.
  • The techniques used in the article are widely used to prepare and analyse nanoparticles/liposomes. The authors may comment on the novelity of the work.

Author Response

Reviewer 1

Author´s answers to comments:

Thank you for your review.

The study is about the development of a controlled release formulation for the delivery of bioactive siRNA. Different approaches has been developed to increase the bioactivity of siRNA in the form lipoplexes, polyplexes etc. Since siRNA knockdown with those carriers has only a transient effect, the loading within a depot, which is capable to control the release over longer time periods for long lasting therapeutic effects, is a desirable scenario. We evaluated three different carriers for their potential as gene delivery vehicles in a GFP DLD1 cell model. The best of these carrier, the DL were subsequently loaded in a drug depot formulation, consisting of a gellan gum hydrogel. The hydrogel was investigated for its drug load, siRNA stability after loading, homogeneity within the hydrogel and the release of lipoplex.

To our knowledge, only few groups have worked on the formulation of loading bioactive siRNA carriers in drug depots, and none of them described the controlled release of lipoplexes from gellan gum hydrogels. We could sustain the release of lipoplexes over 60 days. This provides the basis for a long lasting release of bioactive siRNA, bringing the therapeutic application of depot formulations one step closer to its intended in-vivo use.

Reviewer 2 Report

The presented paper is devoted to very interesting and important topic - controlled delivery of siRNA for gene knockdown therapy. In my opinion paper could be published after corrections.

My major comments are following:

  1. Authors should describe in the introduction and discuss in the results and discussion sections the strategy of application of designed systems. How such system will be introduced into the organism? Prolonged release means that hydrogel should be localised somewhere in the diseased tissue. How this could be realised? Please work on this and describe the strategy. This will raise up the interest to the paper.
  2. The release of siRNA from the composite hydrogel is unclear. The release of lipoplex from hydrogel with subsequent penetration into the cells makes sense, but how can you be sure that you have release of lipoplex, but not of uncomplexed siRNA, which can hardly penetrate the cells interior? This point is unclear from the presented results and discussion?
  3. The morphology of obtained particulate formulations by TEM should be provided. By the way, it is possible to check the medium after release for the presence of lipoplexes by TEM. The distribution of the particles within the hydrogel should be evaluated.

Minor comments:

  1. Page 2, lines 91-93. The aim should br more detailed and strategy description is needed.
  2. Page 3, line 129. It looks like the prepared nanocarriers were mixed with siRNA. But earlier it was stated that they were added upon particles preparation.
  3. Page 5, lines 211-215. The procedure do not allows to distinguish the release of naked and complexed siRNA.
  4. Page 6, line 238. Fig.1 appears first, but Table 1 mentioned first. This should be corrected.
  5. Page 6, line 237. the overall strategy description is lacking for better understanding.
  6. Page 7, Table 1. N/P ratio affects the size. At which N/P the particles were prepared? Please add info to the heading. Why sized for other N/P were nor provided?
  7. Page 7, Figure 2. The toxicity was tested with cancer cells. What about toxicity for cell cultures mimicking normal cells? 
  8. Page 14, line 421. How can you distinguish the release of siRNA and release of lipoplex?

Author Response

  1. Authors should describe in the introduction and discuss in the results and discussion sections the strategy of application of designed systems. How such a system will be introduced into the organism? Prolonged release means that hydrogel should be localised somewhere in the diseased tissue. How this could be realised? Please work on this and describe the strategy. This will raise up the interest to the paper.

ANSWER

The following part was also included in the paper:

Because of these properties, gellan gum offers a wide range of applicability as therapeutic drug depots. Gellan gum hydrogels are stable over a large pH range, physiologically compatible, not cytotoxic and show good shear thinning properties allowing the injection in a syringe [30]. Gellan gum is also investigated for its in-situ gelling behavior [28], increasing the potential of administration even further. Besides the external application for ocular, dermal, or vaginal diseases, the syringeability would allow the application for subcutaneous, intra-articular, or intratumoral indications. Because of these promising attributes, our study aimed to develop a siRNA-loaded gellan gum hydrogel for controlled release applications.

2. The release of siRNA from the composite hydrogel is unclear. The release of lipoplex from hydrogel with subsequent penetration into the cells makes sense, but how can you be sure that you have release of lipoplex, but not of uncomplexed siRNA, which can hardly penetrate the cells interior? This point is unclear from the presented results and discussion?

ANSWER

We thank the referee for this comment. Because we did see biological activity (GFP knockdown) of the release material, we conclude on the release of complexed siRNA. Uncomplexed siRNA which would, as you also state in your comment, hardly penetrate into the cells.

  1. The morphology of obtained particulate formulations by TEM should be provided. By the way, it is possible to check the medium after release for the presence of lipoplexes by TEM. The distribution of the particles within the hydrogel should be evaluated.

ANSWER

Please find the TEM images of the formulations included in the Supplementaries. It was not possible to check the media for lipoplexes by transmission electron microscopy, since the concentration was too low and the differentiation between lipoplex and gel degradation products in the media is not possible. The distribution in the hydrogel is evaluated by Figure 3 C, which shows the fluorescence distribution of labeled siRNA-Cy3 within the hydrogel.

Minor comments:

  1. Page 2, lines 91-93. The aim should br more detailed and strategy description is needed.

ANSWER

Because of the promising attributes (mentioned in the text), our study aimed to develop a siRNA-loaded gellan gum hydrogel for controlled release applications. The development can be separated into three parts in which the carrier finding and characterization is part I, the drug loading and hydrogel formulation is part II and the controlled release is part III.

2. Page 3, line 129. It looks like the prepared nanocarriers were mixed with siRNA. But earlier it was stated that they were added upon particles preparation.

ANSWER

Carrier were produced and afterward mixed in the respective ratio with siRNA before the experiment. Since chitosan polyplexes were freshly prepared before each experiment, siRNA was added directly.

3. Page 5, lines 211-215. The procedure do not allows to distinguish the release of naked and complexed siRNA.

ANSWER

Separation of naked siRNA and complexed siRNA is not possible by this procedure, see also at point 2 of major revision and point 8 of minor revision for the discussion.

4. Page 6, line 238. Fig.1 appears first, but Table 1 mentioned first. This should be corrected.

ANSWER

Corrected, thank you.

5. Page 6, line 237. the overall strategy description is lacking for better understanding.

ANSWER

Thank you for pointing this out. The strategy is explained clearer now on page 2, lines 93-96.

6. Page 7, Table 1. N/P ratio affects the size. At which N/P the particles were prepared? Please add info to the heading. Why sized for other N/P were nor provided?

ANSWER

N/P ratio was determined to be 1:1 for DL, 20:1 for CP, and 10:1 for SLC, based on the results received from gel electrophoresis. At these respective ratios, the siRNA was completely bound in excess to the carrier. Since N/P ratios determine also the transfection efficiency and cell toxicity as well, the other ratios will be part of future experiments. However, this paper was supposed to focus on controlled release of a bioactive carrier.

7. Page 7, Figure 2. The toxicity was tested with cancer cells. What about toxicity for cell cultures mimicking normal cells? 

ANSWER

DLD1 cells were chosen as reporter model for the transfection as well as a hypothetical model for possible clinical uses of tumors. The effect of siRNA needed to be distinguished from possible cytotoxic effects to prevent biased results. However, we agree on that point, that for the development of a depot formulation, the cytotoxicity for the surrounding environment should be assessed, e.g. for fibroblasts and myoblasts. In further development, other cell cultures will be investigated.

8. Page 14, line 421. How can you distinguish the release of siRNA and release of lipoplex?

ANSWER

The sentence in 421 was misleading, this was corrected in the text. The comparison between HPLC and fluorescence intensity was not made to state that both methods are applicable for measuring the release of lipoplex-siRNACy3. The statement was that release measured by fluorescence intensity covers appropriately how quickly siRNA is released from the hydrogel. Even though the method does not allow for the separation of naked siRNA and complexed siRNA, there are points indicating that siRNA was partly released in the form of lipoplexes. Please check also the answer to the major point 1 of your review. Other analytical methods that may allow distinguishing between complexed and naked siRNA, are the following:

  1. Gel electrophoresis with decomplexation e.g. by heparin. However a higher sample concentration is required for the application of low volumes in the gel molds, however, sample processing (evaporation of solvent and decomplexation) may destroy analyte
  2. Transmission electron microscopy, however, sample concentration is not high enough in the release media and the complex environment disturbs the separation of lipoplexes from the background, e.g. gel degradation products.

We thank the reviewer for these interesting questions and suggestions.

Round 2

Reviewer 2 Report

Authors have met all mentioned issues and did their best to improve the MS.

In my opinion the paper could be accepted for publication.